# Thermostable and O_2_-Insensitive Pyruvate Decarboxylases from Thermoacidophilic Archaea Catalyzing the Production of Acetaldehyde

**DOI:** 10.3390/biology11081247

**Published:** 2022-08-22

**Authors:** Faisal Alharbi, Thomas Knura, Bettina Siebers, Kesen Ma

**Affiliations:** 1Department of Biology, University of Waterloo, Waterloo, ON N2L3G1, Canada; 2Molecular Enzyme Technology and Biochemistry (MEB), Environmental Microbiology and Biotechnology (EMB), Centre for Water and Environmental Research (CWE), Faculty of Chemistry, University of Duisburg-Essen, 45141 Essen, Germany

**Keywords:** ethanol fermentation, pyruvate decarboxylase, pyruvate:ferredoxin oxidoreductase, archaea, hyperthermophiles, *Sulfolobus acidocaldarius*, *Saccharolobus solfataricus*

## Abstract

**Simple Summary:**

Pyruvate decarboxylase (PDC) is a key enzyme involved in ethanol fermentation, a process for the production of biofuels. Thermostable and oxygen-stable PDC activity is highly desirable for biotechnological applications at high temperatures. The enzymes from the thermoacidophiles *Saccharolobus* (formerly *Sulfolobus*) *solfataricus* (Ss, T_opt_ = 80 °C) and *Sulfolobus acidocaldarius* (Sa, T_opt_ = 80 °C) were purified and characterized, and their biophysical and biochemical properties were determined comparatively. The purified enzymes were CoA-dependent and thermostable. There was no loss of activity in the presence of oxygen. In conclusion, both thermostable SsPDC and SaPDC catalyze the CoA-dependent production of acetaldehyde from pyruvate in the presence of oxygen.

**Abstract:**

Pyruvate decarboxylase (PDC) is a key enzyme involved in ethanol fermentation, and it catalyzes the decarboxylation of pyruvate to acetaldehyde and CO_2_. Bifunctional PORs/PDCs that also have additional pyruvate:ferredoxin oxidoreductase (POR) activity are found in hyperthermophiles, and they are mostly oxygen-sensitive and CoA-dependent. Thermostable and oxygen-stable PDC activity is highly desirable for biotechnological applications. The enzymes from the thermoacidophiles *Saccharolobus* (formerly *Sulfolobus*) *solfataricus* (Ss, T_opt_ = 80 °C) and *Sulfolobus acidocaldarius* (Sa, T_opt_ = 80 °C) were purified and characterized, and their biophysical and biochemical properties were determined comparatively. Both enzymes were shown to be heterodimeric, and their two subunits were determined by SDS-PAGE to be 37 ± 3 kDa and 65 ± 2 kDa, respectively. The purified enzymes from *S. solfataricus* and *S. acidocaldarius* showed both PDC and POR activities which were CoA-dependent, and they were thermostable with half-life times of 2.9 ± 1 and 1.1 ± 1 h at 80 °C, respectively. There was no loss of activity in the presence of oxygen. Optimal pH values for their PDC and POR activity were determined to be 7.9 and 8.6, respectively. In conclusion, both thermostable SsPOR/PDC and SaPOR/PDC catalyze the CoA-dependent production of acetaldehyde from pyruvate in the presence of oxygen.

## 1. Introduction

Ethanol fermentation using sugars as substrates involves the central metabolic pathways of carbohydrates, in which pyruvate is a key intermediate that is then converted to ethanol by two different pathways (Figure 1) [1,2,3]. In organisms such as *Saccharomyces cerevisiae* and *Zymomonas mobilis*, ethanol is produced by a two-step metabolic pathway [3,4] which is catalyzed by pyruvate decarboxylase (PDC) and alcohol dehydrogenase (ADH), respectively [3]. It is observed that when pyruvate is added, the ratio of carbon partitioning of ethanol increases [5]. In clostridia and many thermophilic microorganisms, pyruvate is converted to ethanol using a three-step pathway [6]. The acetyl-CoA production from pyruvate is catalyzed by pyruvate:ferredoxin oxidoreductase (POR) or pyruvate formate lyase (PFL) [7,8], which is then converted to acetaldehyde by a CoA-dependent acetaldehyde dehydrogenase (AcDH). The conversion of acetaldehyde to ethanol is catalyzed by ADH [8,9].

Commonly known PDCs have been found in many organisms including prokaryotes, such as *Z. mobilis*, and *Sarcina ventriculi* [10,11], and eukaryotes, such as *S. cerevisiae*, sweet potato, wheat, cottonwood, and fish species like carp and goldfish [12]. PDC is a tetrameric enzyme that consists of four identical or non-identical subunits with a molecular mass of approximate 60 kDa [13]. However, no homolog of such PDC has been found in hyperthermophilic bacteria or archaea [3]. Hence, they appear to utilize a bifunctional POR/PDC for catalyzing the decarboxylation of pyruvate to acetaldehyde [14], which was first reported in the anaerobic archaeon *Pyrococcus furiosus* [15]. This new type of POR/PDC is also present in other anaerobic hyperthermophilic archaea such as *Thermococcus guaymasensis* and bacteria like *Thermotoga maritima* and *Thermotoga hypogea*, which have the ability to catalyze both non-oxidative and oxidative decarboxylation of pyruvate to produce either acetaldehyde or acetyl-CoA, respectively [8,14,15]. These bifunctional POR/PDCs are oxygen sensitive and CoA dependent, and apparently CoA only has a structural role in the PDC catalysis [15]. The difference between bifunctional POR/PDC and commonly known PDC can be shown in terms of oxygen sensitivity, dependency on CoA, and lower catalytic activity [8,16]. It is reported that the PDC activity from *Sulfurisphaera* (formerly *Sulfolobus*) *tokodaii* POR/PDC is not oxygen sensitive and that PDC activity increased by approximately 20% in the presence of CoA [16]. However, its catalytic properties such as optimal pH and thermal activity have yet to be determined. Another type of bifunctional POR/PDC was characterized from the hyperthermophilic, anaerobic bacterium *T. maritima* [17], which possesses additional acetohydroxyacid synthase (AHAS) activity catalyzing the production of acetolactate from pyruvate [18].

Most bifunctional POR/PDCs are oxygen sensitive, and the reason for that is suggested to be the oxygen-sensitive FeS-cluster structure of the enzyme [3,19]. It is found that their exposure to air causes the conversion of a stable [4Fe-4S]^2+^ state to an unstable [4Fe-4S]^3+^ form which inactivates the enzyme [19,20]. Some PORs are found to be stable in the presence of oxygen such as those from *Desulfovibrio africanus*, *Halobacterium salinarium*, *S. tokodaii*, and *S. solfataricus* [19,21,22,23]. These oxygen-insensitive enzymes are heterodimers (αβ-type) and possess only one [4Fe-4S] cluster and do not have a δ subunit/domain or dicluster-type ferredoxin carrying two [4Fe-4S] clusters, which is present in homodimeric and heterotetrameric POR/PDCs, respectively [20,22,24].

Although *Sulfolobus*/*Saccharolobus* species are similar in optimal pH, they are different in growth on various substrates, thus differing in metabolic versatility [25,26,27,28,29,30,31]. *S. solfataricus* grows on peptides, amino acids, and sugars including pentoses, hexoses, and polysaccharides [26,29,30,32]. In contrast, *S. acidocaldarius* grows on a wide range of amino acids but a limited range of sugars such as sucrose, maltotriose, dextrin, starch, D-glucose, D-fucose, D-xylose, and L-arabinose [25,26,30,32]. *S. acidocaldarius* shows no diauxic growth on pentoses and hexoses and thus has the ability to grow simultaneously on D-glucose and D-xylose because of the absence of carbon catabolite repression [26,30,32,33]. Growing simultaneously on mixed sugars is preferable for the production of biofuel from cellulosic biomass because there is no need for isolating different sugars in the biomass pre-treatment process [26,32,33]. As a result, the time required for the fermentation of both sugars and the cost of biofuel production would be reduced [32,33].

*Sulfolobus*/*Saccharolobus* POR/PDCs are of special interest for applications compared to other POR/PDCs from hyperthermophiles taking their oxygen resistance and stability into consideration [22,34]. Therefore, the biophysical and biochemical properties of the POR/PDCs from *S. solfataricus* and *S. acidocaldarius* were determined. Here we report the purification and characterization of the thermostable, oxygen-insensitive bifunctional POR/PDCs from *S. acidocaldarius* and *S. solfataricus.*

## 2. Material and Methods

### 2.1. Microorganisms and Chemicals

*S*. *solfataricus* and *S. acidocaldarius* were grown on Brock basal medium complemented with 0.3% (*w*/*v*) NZamine in a 100 L fermenter [30,35]. Acetonitrile was from Fisher Scientific (Whitby, ON, Canada), 2,4-dinitrophenylhydrazine (DNPH) was obtained from Eastman Organic Chemicals (New York, NY, USA), and coenzyme A (CoA) was from US Biological (Salem, MA, USA). Sodium pyruvate, benzyl viologen, methyl viologen, acetaldehyde, dichloromethane, and hydrochloride acid were purchased from Sigma-Aldrich Canada Ltd. (Oakville, ON, Canada). Dithiothreitol (DTT) was purchased from Bio Basic Inc. (Markham, ON, Canada). All other chemicals with high purities were commercially available products unless specified.

### 2.2. Buffer Preparation

For testing the pH effect on enzyme activities, different buffers were used to achieve specific pH values that were measured at room temperature. All pH values presented were those at assay temperatures that were determined based on Δp*K_a_*/°C of each buffer used. These buffers were sodium phosphate (Δp*K_a_*/°C = −0.0028), N-(2-hydroxyethyl)-peperazine-N′-(3-propanedulfonic acid) [EPPS, Δp*K_a_*/°C = −0.015], glycine (Δp*K_a_*/°C = −0.025), and 3-(Cyclohexylamino)-1-propanesulfonic acid [CAPS, Δp*K_a_*/°C = −0.009].

### 2.3. Enzyme Assay

POR activity was determined by measuring the pyruvate- and CoA-dependent reduction of benzyl viologen at 578 nm under anaerobic conditions at 80 °C [15,22]. The assay mixture (2 mL), containing 100 mM sodium phosphate or another specified buffer, 5 mM pyruvate, 1 mM benzyl viologen or methyl viologen, and approximately 50 µM sodium dithionite (SDT) in a glass cuvette with 1 cm light path, was incubated for 4 min to reach 80 °C. After the addition of the enzyme (SsPOR/PDC or SaPOR/PDC), 100 µM CoA was added to start the enzymatic reaction. The absorbance change at 578 nm was recorded using a Genesys 10 Vis spectrophotometer (benzyl viologen ε_578_ = 8.65 mM^−1^ cm^−1^ [36]; methyl viologen ε_578_ = 9.8 mM^−1^ cm^−1^ [37]). One unit of enzyme activity was defined as the oxidation of 1 μmol of pyruvate or the reduction of 2 μmol of benzyl viologen/methyl viologen per min.

The activity of PDC was determined by measuring the acetaldehyde production using the DNPH derivatization method followed by high-performance liquid chromatography (HPLC) [8,15]. Enzymatic reactions were carried out at 80 °C. The assay mixture (1 mL), containing 100 mM sodium phosphate pH 7.9 or another specified buffer, 10 mM pyruvate, and 100 µM CoA in sealed 8 mL vials, was incubated in a water bath (80 °C) before adding the enzyme and then incubated for 2 h. The enzymatic reaction was stopped by transferring the vials to an ice bath followed by adding 2 mL of saturated DNPH solution in 2 N HCl, which derivatizes acetaldehyde, producing a yellow-reddish-colored compound. The vials were shaken in the dark overnight at 315 rpm and room temperature. The extraction of acetaldehyde–DNPH derivative was performed twice by adding 1 mL of dichloromethane (DCM) in the vials each time and followed by shacking for 30 min. The lower organic phase was transferred to a new vial that was then covered by Parafilm M membrane punctured with a needle to create a few small holes and placed in a vacuum desiccator to evaporate the DCM. The resulting yellowish-red powder was dissolved in 4 mL of acetonitrile and incubated at 4 °C overnight and subsequently filtered through 0.45 µm nylon syringe filter (Mandel Scientific, Guelph, ON, Canada). The filtered product was analyzed at room temperature using a Perkin Elmer (Akron, OH, USA) series 4 HPLC system equipped with a reversed-phase Allure C18 5 µm column (150 × 4.6 mm). Samples (80 µL) were injected to the Rheodyne injection valve using a 100 µL micro-syringe. The mobile phase of acetonitrile/water (80:20 *v*/*v*) was used at a flow rate of 1 mL·min^−1^, and the acetaldehyde–DNPH was detected at 365 nm by a micro-metrics model 788 dual variable wavelength detector. The concentration of acetaldehyde was measured based on an acetaldehyde standard curve prepared under the same assay conditions. One unit of activity was defined as the formation of 1 μmol of acetaldehyde per min.

### 2.4. Preparation of Cell-Free Extract

Cell-free extract (CFE) was prepared anaerobically from frozen *S. solfataricus* and *S. acidocaldarius* cells. *S. solfataricus* cell pellets (approximately 5 g, wet weight) were transferred into a degassed serum bottle and suspended in the anaerobic buffer (30 mL) containing 10 mM sodium phosphate and 2 mM dithiothreitol (DTT) at pH 7.0, while *S*. *acidocaldarius* cell pellets (approximately 5 g, wet weight) were transferred into a degassed serum bottle and suspended in the anaerobic buffer (30 mL) containing 50 mM Tris-HCl and 2 mM DTT at pH 7.3. The suspensions were stirred for 2 h at 30 °C. Cell suspensions were run through a French Press Cell (Thermo Scientific, Waltham, MA, USA) four times at 20,000 psi to break the cells. The obtained crude cell extracts were centrifuged anaerobically at 20,000× *g* for 30 min at 4 °C. The supernatants were CFEs that were transferred to anaerobic serum bottles for further use. Protein concentration was determined using the Bradford microassay method, and bovine serum albumin served as protein standard [38].

### 2.5. Purification of Enzymes

The enzyme purification was carried out at room temperature and under anaerobic conditions. A fast performance liquid chromatography (FPLC) system with a P-920 pump (Amersham Pharmacia Biotech, Baie D’urfe, QC, Canada) was used. POR activity in each fraction was monitored, while PDC activity was measured after the final purification step. SDS-PAGE was used for determining the purity of the fractions according to Laemmli’s method [39].

Prepared CFE of *S. solfataricus* or *S*. *acidocaldarius* was diluted with buffer A (50 mM Tris-HCl pH 7.8 and 2 mM dithiothreitol [DTT]) by a ratio 1:1 (*v*/*v*), and loaded onto a DEAE-Sepharose column (2.6 cm × 11 cm) that was equilibrated with buffer A. The column was washed with one volume of the column (60 mL) using buffer A, then eluted with five volumes of the column (300 mL) linear gradient of (0–1 M) buffer B containing 1 M NaCl, 50 mM Tris-HCl pH 7.8, and 2 mM DTT. The flow rate was 2 mL·min^−1^. The fractions with enzyme activities (33–75 mM NaCl for *S. solfataricus* and 230–300 mM NaCl for *S. acidocaldarius*) were loaded onto a hydroxyapatite column (HAP, 2.6 cm × 9 cm) equilibrated with buffer A at a flow rate of 1 mL min^−1^. After washing with buffer A (one column volume (50 mL)), proteins bounded to the column were eluted with a linear gradient of 0–0.5 M potassium phosphate using buffer A and buffer C containing 0.5 M potassium phosphate, 50 mM Tris-HCl pH 7.8, and 2 mM DTT. Fractions containing high POR activity (125–310 mM potassium phosphate for *S. solfataricus* and 280–350 mM potassium phosphate for *S. acidocaldarius*) were pooled and loaded onto a phenyl-Sepharose column (2.6 cm × 12 cm) equilibrated with buffer D containing 0.5 M ammonium sulfate in buffer A. After washing the column using buffer D (65 mL) at a flow rate of 2 mL min^−1^, the enzyme was eluted out by applying a reverse linear gradient of 0.5 M to 0.0 M ammonium sulfate. Fractions revealing POR activity for *S. solfataricus* (110–55 mM ammonium sulfate) or *S*. *acidocaldarius* (165–110 mM ammonium sulfate) were combined, washed, and concentrated with ultrafiltration using 30 kDa high-flow, polyethersulfone (PES) membranes (Sigma-Aldrich, Oakville, ON, Canada).

## 3. Results

### 3.1. Enzyme Purification

CFEs from *S*. *solfataricus* and *S. acidocaldarius* were prepared for the enzyme purification. The *S*. *solfataricus* PDC and POR activities of the bifunctional POR/PDC present in its CFE were determined to be 0.0027 ± 0.0003 U/mg and 0.18 ± 0.01 U/mg, respectively. Similarly, the *S*. *acidocaldarius* PDC and POR activities present in CFE were determined to be 0.0011 ± 0.0004 U/mg and 0.10 ± 0.01 U/mg, respectively. Due to using a much simpler assay method, only POR activities were followed for the purification of the enzymes. The enzyme from *S*. *solfataricus* was purified approximately 42-fold with a recovery of 25% (Table 1), while the enzyme from *S*. *acidocaldarius* was purified approximately 70-fold with a recovery of 19% (Table 2). SDS-PAGE was performed to estimate the purity of enzyme-containing fractions after each column, showing both purified enzymes (~80–90% purity) are heterodimeric with subunit molecular masses of 37 ± 3 kDa and 66 ± 2 kDa, respectively (Figure 2)

### 3.2. O_2_-Sensitivity and Thermostability of the Purified Enzymes

The enzymes were purified under anaerobic conditions. When exposed to air/oxygen for 7 h at 4 °C, the PDC activities of purified SsPOR/PDC and SaPOR/PDC of 0.1 ± 0.01 U/mg and 0.031 ± 0.005 U/mg remained to be 0.083 ± 0.007 U/mg and 0.025 ± 0.002 U/mg, respectively. Upon exposure for 48 h at 4 °C, the SsPOR and SaPOR activities of 4.6 ± 0.2 U/mg and 5.5 ± 0.05 U/mg remained to be 4.6 ± 0.1 U/mg and 5.3 ± 0.2 U/mg, respectively. The results showed that both the PDC and POR activities of the bifunctional enzymes were not oxygen sensitive.

Both the PDC and POR activities of the purified enzymes increased along with the increase of assay temperature from 30° to 80 °C, and the optimal temperatures for SsPDC, SaPOR, SaPDC, and SsPOR activity were found to be 80 °C, 80 °C, >90 °C, and >90 °C, respectively (Figure 3A,B). There were no enzyme assays performed at temperatures higher than 90 °C because of technical difficulties. The thermostabilities of the enzymes were determined by monitoring their residual POR activities after incubation for different time intervals at different temperatures. The half-life time required for losing 50% of *S*. *solfataricus* enzyme activity (t_1/2_) was found to be approximately 5.5 h at 70 °C and 2.9 h at 80 °C. The half-life time for the *S*. *acidocaldarius* enzyme activity was determined to be approximately 6.4 h at 70 °C and 1.1 h at 80 °C.

The activation energy (E_act_) for SsPOR and SaPOR activity, as calculated from the Arrhenius plots over the range of 60–90 °C (Figure 3A,B), was found to be 33.2 kJ/mol and 47 kJ/mol, respectively, while the activation energy for SsPDC and SaPDC activity (over the range of 50–90 °C) was 44 kJ/mol and 70 kJ/mol, respectively.

### 3.3. Catalytic Properties of the Purified Enzymes

For testing the pH effect on enzyme activities, different buffers were used to achieve specific pH values at the respective assay temperatures. The optimal pH values for both activities, PDC and POR, from the *S*. *solfataricus* and *S. acidocaldarius* enzyme were determined to be pH 7.9 and pH 8.6 at 80 °C, respectively (Figure 4A,B).

To determine the kinetic parameters, the POR and PDC activities of *S. solfataricus* and *S. acidocaldarius* were determined by varying concentrations of pyruvate (0–10 mM) and CoA (0–100 µM) at 80 °C, and it was found that these activities were dependent on both pyruvate and CoA (Figure 5). The kinetic parameters were then calculated by fitting the Michaelis–Menten equation for pyruvate and CoA (Table 3). The apparent *K*_m_ values of SsPOR and SaPOR activity for pyruvate were 0.5 ± 0.1 mM and 0.3 ± 0.05 mM, respectively, and the apparent *K*_m_ values for CoA were found to be 10.7 ± 0.4 µM and 21.5 ± 0.3 µM, respectively (Table 3). The apparent *K*_m_ values of SsPDC activity for pyruvate and CoA were 1.1 ± 0.2 mM and 0.77 ± 0.27 µM, respectively. The apparent *K*_m_ values of SaPDC activity for pyruvate and CoA were 0.86 ± 0.2 mM and 0.3 ± 0.06 µM, respectively (Table 3). It appeared that the apparent *K*_m_ values for pyruvate were approximately 1 mM; however, the apparent *K*_m_ values for CoA were significantly different with 10.7–21.5 µM for POR and 0.3–0.8 µM for PDC activity (Table 3). These significant differences are also visible in Figure 5 and may be related to the fact that CoA serves as a cosubstrate of POR but not for PDC activity [15]. For this reason, the estimated apparent *K*_m_ values for PDC activities may be an indication of the level of CoA requirement for PDC catalytic activities under these assay conditions.

## 4. Discussion

PDC and POR are key enzymes for the production of ethanol from pyruvate using a two-step pathway [3,4] and a three-step pathway, respectively [6] (Figure 1). However, there is no commonly known PDC found in hyperthermophiles. Bifunctional POR/PDCs have been discovered in several hyperthermophilic microorganisms which have the ability to catalyze both oxidative and non-oxidative decarboxylation of pyruvate. A great challenge of these POR/PDCs is their oxygen sensitivity [14,15].

The purification of POR/PDCs from *S**. solfataricus* and *S. acidocaldarius* was carried out successfully, showing that both are heterodimeric enzymes, which is in agreement with the reports of such enzymes from *S**. solfataricus* and *S. tokodaii* [16,20]. Purification was achieved approximately 42-fold for SsPOR/PDC and 70-fold for SaPOR/PDC via three purification steps (Table 1 and Table 2). Although the purities of the purified enzymes are estimated to be approximately 80–90%, there was no indication of any interference in the enzyme assays except that specific activities might be approximately 10% underestimated. The determination of POR and PDC activities of the purified enzymes from *S**. solfataricus* and *S. acidocaldarius* proved that they have bifunctional POR/PDC enzymes similar to other hyperthermophiles [3,15,16].

Our results show that POR/PDCs from *S**. solfataricus* and *S. acidocaldarius* are stable in the presence of air/oxygen. This is consistent with previous reports from *S. tokodaii* demonstrating oxygen-insensitive PDC activity [16]. The mechanism of the oxygen resistance of these enzymes is found to be the lack of other oxygen-sensitive [4Fe-4S] clusters [16]. The removal of the only oxygen resistant [4Fe-4S] cluster from the *S. tokodaii* enzyme inhibited POR activity but had no effect on the PDC activity, indicating that the [4Fe-4S] cluster is required for oxidative decarboxylation but not needed for non-oxidative decarboxylation of pyruvate [16]. Obviously, POR/PDCs from *S**. solfataricus* and *S. acidocaldarius* also showed the remarkable advantageous feature of being active in the presence of oxygen. Furthermore, *S**. solfataricus* PDC activity (0.13 U/mg at 80 °C) is almost twice that (0.07 U/mg at 80 °C) of *S. tokodaii* [16], which warrants further investigation for potential applications in biotechnology.

The highest SsPOR activity was measured at 90 °C, which is similar to POR activities from *P. furiosus*, *T. maritima*, *T. guaymasensis*, and *S. tokodaii* [40]; however, SaPOR activity revealed an optimum temperature of 80 °C. The optimal temperature for SsPDC was determined to be at 80 °C, which is similar to PDCs from *T. hypogea* and *T. guaymasensis* [14]. SaPDC activity increased continuously until 90 °C, which is similar to PDCs from *P. furiosus*, and *T. maritima* [14,15]. Analyses of the thermostability of SsPOR/PDC (following POR activity) revealed a half-life time (t_1/2_) of ~175 min at 80 °C which is similar to POR from *T. hypogea*; on the other hand, SaPOR/PDC showed a half-life time of ~65 min at 80 °C. In contrast, only 60% of POR activity from *S. tokodaii* POR/PDC remains active after incubation for 30 min at 80 °C [34]. It is concluded that the *S. solfataricus* enzyme is, so far, the most thermostable POR/PDC from members of the *Sulfolobales*.

The activation energy (E_act_) values for SsPOR and SaPOR activity were found to be 33.2 kJ/mol and 47 kJ/mol, which are similar to POR from *T. hypogea* (34.8 kJ/mol range of 60–95 °C), while POR from *T. maritima* had a lower E_act_ (23.6 kJ/mol range of 50–80 °C) compared to other PORs [14]. On the other hand, the POR from *A. fulgidus* had an E_act_ of 75 kJ/mol for the range of 30–70 °C [40]. The activation energy for SsPDC and SaPDC activity over the range of 50–90 °C were 44 kJ/mol and 70 kJ/mol, respectively, which are approximately 40% higher than their corresponding POR activities.

POR activity from *S**. solfataricus*, and *S. acidocaldarius* POR/PDCs showed a pH optimum at pH 8.6, which is similar to POR activity from *S. tokodaii* (pH 8.5) [34] and close to the mesophilic bacterium *D. africanus* and the archaeon *H. salinarium* (pH 9.0) [21,41]. The pH optimum for POR activity from *S**. solfataricus* POR/PDC was previously reported to be at pH 7.0–8.0; it is likely that the use of different assay conditions may contribute to the observed difference in optimal pH determination [22]. POR/PDCs from *H. salinarium* and members of the *Sulfolobales* are determined to be highly identical in their enzyme structures [20]. The optimum pH of PDC activity from hyperthermophilic POR/PDCs are reported to be higher than the PDCs from mesophilic organisms, which is consistent with the results for SsPDC and SaPDC in this study (pH 7.9, 80 °C) [14]. In addition, optimal pH values of PDC activities from hyperthermophilic POR/PDCs are reported to be equal to or higher than those of their corresponding POR activities; however, for SsPDC and SaPDC activities, the optimum pH was lower than for their POR activities [14,15]. In general, optimum pHs of hyperthermophilic POR/PDCs are higher than those of the commonly-known PDCs from bacteria and fungi, which prefer slightly acidic environments (approximate pH 5–6).

The bifunctional SsPOR/PDC and SaPOR/PDC showed the capability of catalyzing the oxidative and non-oxidative decarboxylation of pyruvate in the presence of CoA. The apparent *K*_m_ values for pyruvate of SsPOR and SaPOR activity were similar to POR activities from other hyperthermophiles (Table 4). There was no activity detectable when the CoA was omitted from both PDC and POR activity assays (Figure 5). SsPOR and SaPOR activity was found to be CoA dependent, which is similar to POR activities from other members of the *Sulfolobales*. SsPOR activity showed lower apparent *K*_m_ value for CoA (10 µM) than other characterized hyperthermophilic POR activities; however, the apparent *K*_m_ value for CoA of SaPOR activity (18.2 µM) was similar to that of the hyperthermophilic crenarchaeon *S. tokodaii* (17 µM) and the hyperthermophilic bacterium *T. hypogea* (21 µM).

Although it was concluded that PDC activity from *S. tokodaii* POR/PDC was not CoA dependent, results showed that the addition of CoA into the assay mixture actually enhanced the PDC activity by approximately 20% [16], indicating that CoA still plays an important role in PDC catalysis. In this study, the apparent *K*_m_ values for CoA were calculated by fitting the Michaelis–Menten equation to be 0.77 µM for SsPDC and 0.3 µM for SaPDC activity, which appears to be lower than in other PDCs (Table 4). However, it is also known that CoA does not serve as a cosubstrate for PDC activity, and it only plays a structural role for the catalysis [15]. This can also be supported by the observation that the addition of only a small concentration of CoA resulted in a significant increase in PDC activities for both SsPOR/PDC and SaPOR/PDC (Figure 5B,D), indicating the requirement of CoA for achieving their highest PDC activities. It is likely that the calculated apparent *K*_m_ values for PDC activities may merely reflect the level of CoA required for their catalytic activities under those specific assay conditions.

The PDC activities of both SsPOR/PDC and SaPOR/PDC were about 2–3% of the rate of the corresponding POR activities, which is similar to enzymes from hyperthermophilic bacteria, i.e., *T. hypogea* and *T. maritima*, and the thermoacidophilic crenarchaeon *S. tokodaii* [14,16]. However, these rates are much lower than those reported for hyperthermophilic euryarchaeota (about 20%), i.e., *T*. *guaymasensis* [8] and *P. furiosus* [15]. The specific PDC activity of POR/PDCs from members of the *Sulfolobales* (0.03–0.13 U/mg) were found to be lower than the PDC activities from hyperthermophilic euryarchaeota (1.3–4.3 U/mg [8,15]), which may reflect the difference in structure of their corresponding catalytic sites. Although the ratio of PDC to POR activity was similar, SsPOR and SaPOR activities (7.5 U/mg and 7 U/mg, respectively, 80 °C) were much higher than the POR activity (3.6 U/mg, 80 °C) from *S. tokodaii* [16]. In addition, SsPDC activity (0.12 U/mg, 80 °C) was much higher than PDC activity from other members of the *Sulfolobales* (0.04–0.07 U/mg, 80 °C [16], Table 4. Therefore, the CoA-dependent PDC activity is similar to other hyperthermophilic POR/PDCs, indicating a structural role of CoA required in the catalysis of the non-oxidative decarboxylation of pyruvate [15].

## 5. Conclusions

The purification and characterization of PDCs/PORs from *S**. solfataricus* and *S. acidocaldarius* has proved that they are bifunctional, thermostable and oxygen insensitive enzymes. It has been determined that SsPDC and SaPDC activities are CoA-dependent and that their apparent *K*_m_ values are much lower than those of other characterized hyperthermophilic PDCs. The characterization of SsPOR/PDC and SaPOR/PDC have been accomplished for the first time, including the kinetic parameters, pH and temperature optimum. These results provide further insight into the kinetic properties of hyperthermophilic POR/PDCs. The oxygen resistant POR/PDCs from members of the *Sulfolobales* may allow for novel strain design and production strategies at high temperatures with higher fermentation efficiency and lower production costs.

## Figures and Tables

**Figure 1 biology-11-01247-f001:**
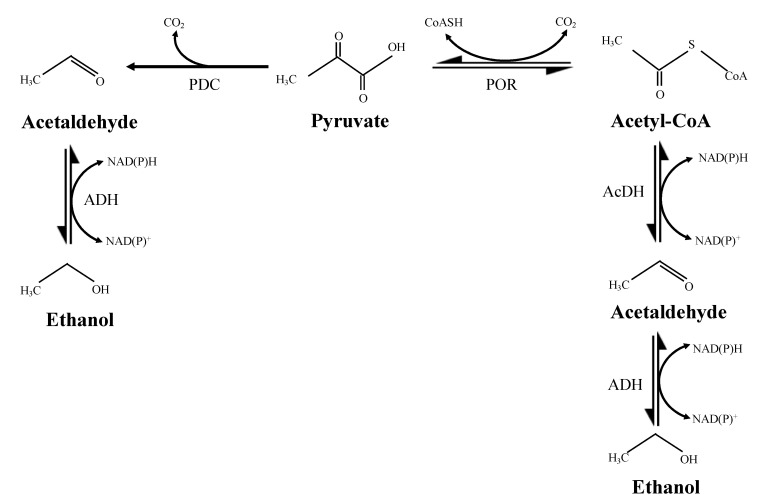
Pathways for ethanol production from pyruvate. Left side, two-step pathways catalyzed by PDC (pyruvate decarboxylase) and ADH (alcohol dehydrogenase); Right side, three-steps pathway catalyzed by POR (pyruvate:ferredoxin oxidoreductase), AcDH (CoA-dependent acetaldehyde dehydrogenase), and ADH (alcohol dehydrogenase).

**Figure 2 biology-11-01247-f002:**
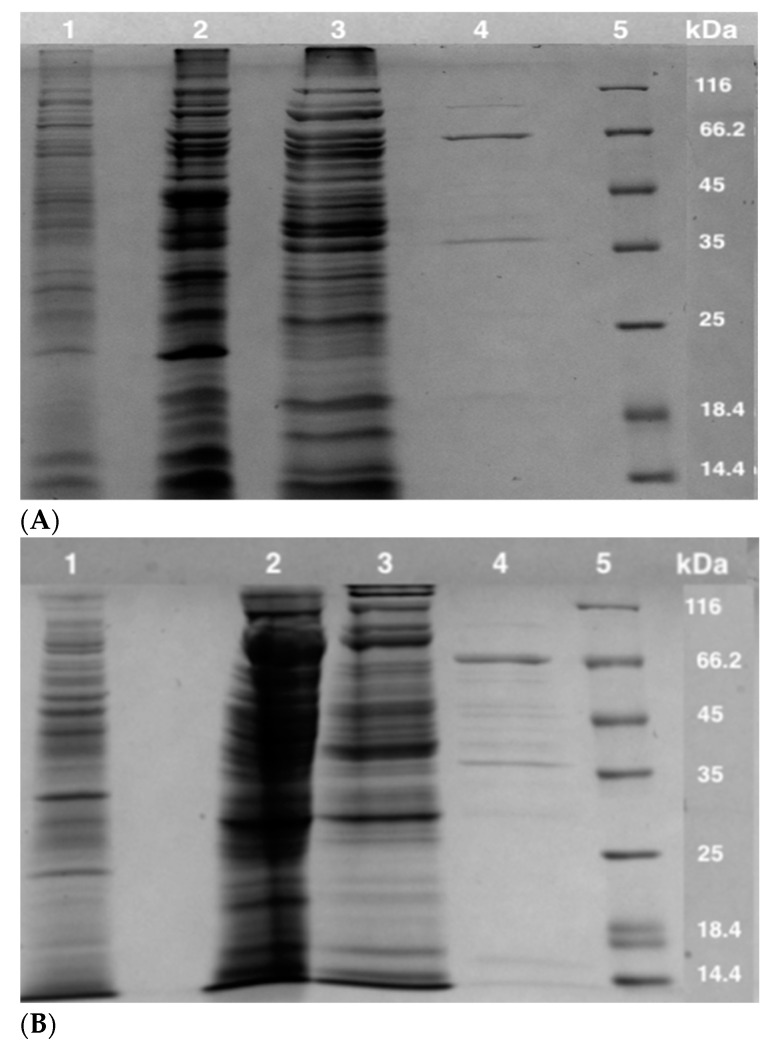
Analyses of purified bifunctional POR/PDCs from *S. solfataricus* (**A**) and *S. acidocaldarius* (**B**) using SDS-PAGE (12.5%). (**A**) Lane 1, 10 μg of CFE; lane 2, 20 µg of DEAE fraction; lane 3, 18 µg of HAP fraction; lane 4, 1.5 µg of purified enzyme; lane 5, BLUeye pre-stained protein ladder. (**B**) Lane 1, 12 μg of CFE; lane 2, 33 µg of DEAE fraction; lane 3, 22 µg of HAP fraction; lane 4, 0.8 µg of purified enzyme; lane 5, BLUeye pre-stained protein ladder.

**Figure 3 biology-11-01247-f003:**
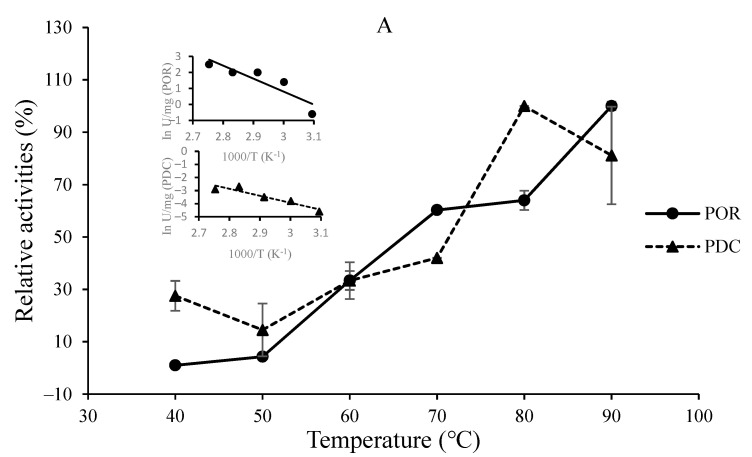
Temperature dependence of the POR and PDC activities of the bifunctional POR/PDCs from *S. solfataricus* (**A**) and *S. acidocaldarius* (**B**). POR and PDC enzyme activities were determined over a temperature range from 40 to 90 °C. The assay mixture of POR contains 100 mM sodium phosphate pH 8.0, 5 mM pyruvate, 100 µM CoA, 1 mM benzyl viologen and approximately 50 µM sodium dithionite. The PDC assay mixture was 100 mM sodium phosphate at pH 8.0, 10 mM pyruvate, and 100 µM CoA. The relative activities of 100% are equal to the highest specific activities (12.1 U/mg for SsPOR, 0.069 U/mg for SsPDC, 8 U/mg for SaPOR, and 0.057 U/mg for SaPDC activities). The insets show the Arrhenius plots.

**Figure 4 biology-11-01247-f004:**
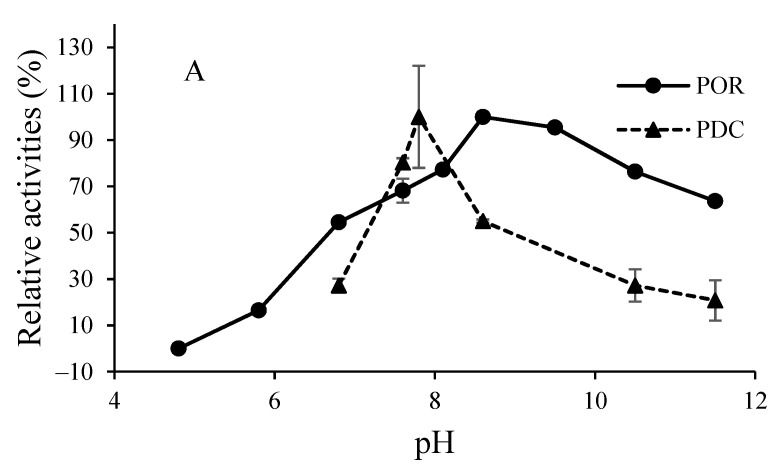
pH dependence of the POR and PDC activities of the bifunctional POR/PDCs from *S. solfataricus* (**A**) and *S. acidocaldarius* (**B**). POR activities were assayed using 5 mM pyruvate, 1 mM methyl viologen, 100 µM CoA, and approximately 50 µM sodium dithionite at 80 °C. The following buffers (100 mM) were used: sodium phosphate (pH 4.9, 5.9, and 6.9), glycine (pH 7.6, 8.1 and 8.6), and CAPS (pH 9.5, 10.5, and 11.5). PDC activities were measured using 10 mM pyruvate and 100 µM CoA at 80 °C. The following buffers were used: sodium phosphate (pH 6.9, and 7.9), glycine (pH 7.6, and 8.6), and CAPS (pH 10.5, and 11.5). The relative activities of 100% are equal to the highest specific activities (2.2 U/mg for SsPOR, 0.16 U/mg for SsPDC, 0.55 U/mg for SaPOR and 0.052 U/mg for SaPDC activity).

**Figure 5 biology-11-01247-f005:**
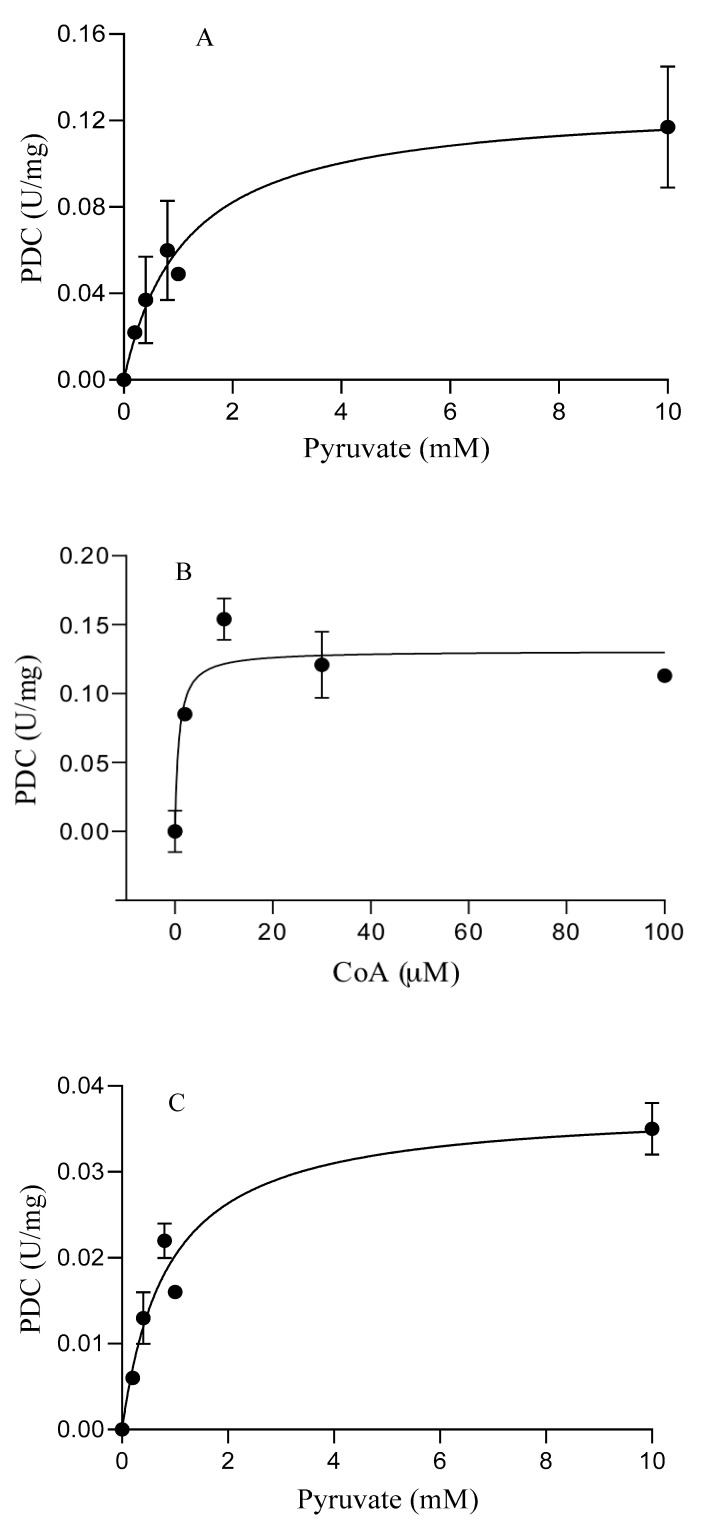
The pyruvate and CoA dependency of the PDC activities of the bifunctional POR/PDCs from *S. solfataricus* (**A**,**B**) and *S. acidocaldarius* (**C**,**D**). The CoA dependency ((**A**,**C**) 0.0 to 100 µM in the presence of 100 mM sodium phosphate [pH 7.9] and 10 mM pyruvate) and pyruvate dependency ((**B**,**D**) 0.0 to 10 mM in the presence of 100 mM sodium phosphate [pH 7.9] and 100 µM CoA) were performed at 80 °C.

**Table 1 biology-11-01247-t001:** Purification of the bifunctional POR/PDC from *S. solfataricus*.

Step	Enzyme	Protein (mg)	Specific Activity (U/mg) ^a,b^	Total Activity (U)	Fold	Recovery (%)
CFE	POR	354.2 ± 2.5	0.18 ± 0.01	63.8 ± 0.15	1	100
	PDC	354.2 ± 2.5	0.0027 ± 0.0003	0.95 ± 0.05	1	100
DEAE	POR	90 ± 1	0.43 ± 0.02	39 ± 0.1	2.4	61.4
	PDC	nd	nd	nd	nd	nd
HAP	POR	63.1 ± 0.5	0.6 ± 0.02	37.9 ± 0.05	3.3	59.3
	PDC	nd	nd	nd	nd	nd
Phenyl-Sepharose	POR	2.1 ± 0.1	7.5 ± 0.05	15.9 ± 0.01	41.6	25
	PDC	2.1 ± 0.1	0.11 ± 0.004	0.23 ± 0.005	40.7	24

^a^ One unit of the POR activity was defined as micromole of pyruvate oxidized per min. ^b^ One unit was defined as the production of 1 μmol of acetaldehyde per min. nd, not determined.

**Table 2 biology-11-01247-t002:** Purification of the bifunctional POR/PDC from *S. acidocaldarius*.

Step	Enzyme	Protein (mg)	Specific Activity (U/mg) ^a,b^	Total Activity (U)	Fold	Recovery (%)
CFE	POR	231.8 ± 2	0.1 ± 0.01	23.2 ± 1	1	100
	PDC	231.8 ± 2	0.0011 ± 0.0004	0.25 ± 0.04	1	100
DEAE	POR	54.23 ± 1.5	0.28 ± 0.01	15.2 ± 0.3	2.38	65.4
	PDC	nd	nd	nd	nd	nd
HAP	POR	22.6 ± 0.4	0.45 ± 0.03	10 ± 0.2	4.5	39.4
	PDC	nd	nd	nd	nd	nd
Phenyl-Sepharose	POR	0.63 ± 0.03	7 ± 0.02	4.41 ± 0.01	70	19
	PDC	0.63 ± 0.03	0.055 ± 0.003	0.035 ± 0.001	50	14

^a^ One unit of the POR activity was defined as micromole of pyruvate oxidized per min. ^b^ One unit was defined as the production of 1 μmol of acetaldehyde per min. nd, not determined.

**Table 3 biology-11-01247-t003:** Kinetic parameters of the POR and PDC activity of the bifunctional POR/PDCs from *S. solfataricus* and *S. acidocaldarius*.

Enzyme Sources	Enzyme Activity	^a^ Pyruvate		^b^ CoA	
		*K*_m_ (mM)	V_max_ (U/mg^−1^)	*K*_m_ (µM)	V_max_ (U/mg^−1^)
*S. solfataricus*	POR	0.5 ± 0.1	6.3 ± 0.7	10.7 ± 0.4	7.7 ± 0.07
	PDC	1.1 ± 0.2	0.12 ± 0.09	0.77 ± 0.27 ^c^	0.12 ± 0.08 ^c^
*S. acidocaldarius*	POR	0.3 ± 0.05	1.9 ± 0.2	21.5 ± 3	1.7 ± 0.08
	PDC	0.86 ± 0.2	0.04 ± 0.03	0.3 ± 0.06 ^c^	0.04 ± 0.03 ^c^

^a^ POR activity was measured using 0.1 mM CoA, 1 mM benzyl viologen 50 µM sodium dithionite, 3 µg protein for SsPOR, and 12 µg protein for SaPOR at 80 °C; and for PDC, 0.1 mM CoA, 25 µg protein for SsPOR/PDC, and 50 µg protein for SaPOR/PDC at 80 °C. ^b^ POR activity was measured using 5 mM pyruvate, 1 mM benzyl viologen, 50 µM sodium dithionite, 3 µg protein for SsPOR, and 12 µg protein for SaPOR at 80 °C; and for PDC, 10 mM pyruvate, 25 µg protein for SsPOR/PDC, and 50 µg protein for SaPOR/PDC at 80 °C. ^c^ Estimated based on the plots (Figure 5B,D).

**Table 4 biology-11-01247-t004:** Kinetic parameters of the POR and PDC activity of bifunctional POR/PDCs from hyperthermophiles.

Organism (Growth T_opt_, °C)		Enzyme Activity (80 °C)	Pyruvate		CoA		References
			*K*_m_ (mM)	V_max_ (U/mg)	*K*_m_ (µM)	V_max_ (U/mg)	
Bacteria	*T. maritima* (80)	POR	0.4 ± 0.1	81 ± 6	63 ± 6	94 ± 2	[14]
		PDC	0.92 ± 0.3	1.4 ± 0.04	3.1 ± 1.2	1.3 ± 0.03	
	*T. hypogea* (70 ^a^)	POR	0.13 ± 0.03	99 ± 3	21 ± 2	73 ± 4	[14]
		PDC	1.4 ± 0.4	2.5 ± 0.18	1.4 ± 0.02	1.6 ± 0.13	
Archaea	*T. guaymasensis* (88)	POR	0.53 ± 0.03	18 ± 0.23	70 ± 10	21.8 ± 0.8	[8]
		PDC	0.25 ± 0.05	3.8 ± 0.14	20 ± 1	3.3 ± 0.09	
	*P. furiosus* (100)	POR	0.46	23.6	110	22	[15]
		PDC ^b^	1.1	4.3 ± 0.3	110	4.3 ± 0.3	
	*S. solfataricus* (80)	POR ^c^	0.5 ± 0.1	6.3 ± 0.7	10.7 ± 0.4	7.7 ± 0.07	This study
		PDC ^d^	1.1 ± 0.2	0.12 ± 0.09	0.77 ± 0.27	0.12 ± 0.08	
	*S. acidocaldarius* (80)	POR ^c^	0.3 ± 0.05	1.9 ± 0.2	21.5 ± 3	1.7 ± 0.08	This study
		PDC ^d^	0.86 ± 0.2	0.04 ± 0.03	0.3 ± 0.06	0.04 ± 0.03	

For POR assays, one unit was defined as 1 μmol of pyruvate oxidized or the reduction of 2 μmol methyl viologen per min at 80 °C and pH 8.4. For PDC assays, one unit was defined as the production of 1 μmol of acetaldehyde per min at 80 °C and pH 8.4. ^a^ Capable of growing at 90 °C. ^b^ One unit was defined as the production of 1 μmol of acetaldehyde per min at 80 °C and pH 10.2. ^c^ One unit was defined as 1 μmol of pyruvate oxidized or the reduction of 2 μmol benzyl viologen per min at 80 °C and pH 8.0. ^d^ One unit was defined as the production of 1 μmol of acetaldehyde per min at 80 °C and pH 7.9.

## Data Availability

Not applicable.

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
