# Peer review of "Thermostable and O_2_-Insensitive Pyruvate Decarboxylases from Thermoacidophilic Archaea Catalyzing the Production of Acetaldehyde"

_biology, 2022, doi:10.3390/biology11081247_

Round 1

Reviewer 1 Report

In this manuscript, Alharbi et al. have investigated biochemical characterization of two pyruvate decarboxylases from the thermoacidophiles Saccharolobus solfataricus and Sulfolobus acidocaldarius. Although the optimal temperature and pH, and kinetic analyses of these two enzymes have been determined, effects of divalent metal ion should be also investigated. I recommended major revision based on my following comments.

Major comments:

1. Since Saccharolobus solfataricus and Sulfolobus acidocaldarius are aerobic, why two enzymes are O2-sensitive? Have the authors determined these enzyme activities in the presence of O2. Additionally, how did the authors purify the enzyme proteins in the absence of O2?

2. As shown in Fig. 1, it seems to me that the two enzyme proteins are not pure since other protein bands appear in lane 4. Furthermore, the loaded proteins are too low, which might confirm the protein impurity.

3. As shown in Fig. 2, it seems to me that the enzyme activities did not increase as increasing tempeartures. I am wondering how the activation energy (Eact) for SsPOR and SaPOR activity was calculated.

4. In the Table 4, the Km values of PDC for CoA substrate 0.77±0.77 and 0.3±0.6. Are the average values higher than their standard deviation values?

Minor comments:

1. The formats of A and B in Figs. 1, 2 and 3 should be uniform.

2. Red wave lines in Figs. 4 and 5 should be removed.

3. Reference 28 is bold.

4. Reference 30 is underlined.  

Author Response

The authors appreciate your reviewing of the manuscript and comments/suggestions, to which the attached file is our response.  The revision has been made accordingly.

Reviewer 2 Report

Authors present a description of the biophysical and biochemical properties of Ss and Sa enzymes respectively. From the SDS-PAGE, considering the low quantity loaded a quite low purity is evident. For a complete characterization did the authors perform a mass spectrometry analysis? 

Authors should make text editing for the following sentences:

#line 67: correct the word "oxgen" with oxygen

#lines 368 and 369: correct the word "half-live" with half-life

#line 371: correct the word "S.solfatricus" with S.solfataricus

#line 489 (in the references): correct the word "oyruvate" with pyruvate

and generally pay attention to any other typing error.

Author Response

(The authors gave the same response as above.)

Reviewer 3 Report

Lane 67 replace oxgen with oxygen

Lane 185 replace Aldrish with Aldrich

I suggest inserting the legend (i.e. S.solfataricus filled circle POR, etc…) inside the graph box, both for figure 2 A and B and for figure 3 A and B

Figure 4 is missed. The authors described in the figure legend “Pyruvate and CoA dependency of PDC activities of…” are instead indicated the Pathways for ethanol production from pyruvate

Moreover, Figure 5, describing the Pathways for ethanol production from pyruvate, should be placed at the beginning of the manuscript as Figure 1

If possible the authors should show a polyacrylamide gel showing the co-purification of the two subunits in the supplementary materials

The font size of the figures and table legends must be uniform

Author Response

(The authors gave the same response as above.)
